METHODS

# SPIN-CGNN: Improved fixed backbone protein design with contact map-based graph construction and contact graph neural network

**Xing Zhang**[1,2], **Hongmei Yin**[2], **Fei Ling**[1]*, **Jian Zhan**[2], **Yaoqi Zhou**[2]

**1** School of Biology and Biological Engineering, South China University of Technology, Guangzhou, People's Republic of China, **2** Institute of Systems and Physical Biology, Shenzhen Bay Laboratory, Shenzhen, People's Republic of China

* fling@scut.edu.cn (FL), zhanjian@szbl.ac.cn (JZ), zhouyq@szbl.ac.cn (YZ)

**Data Availability Statement:** Code Availability: source code and datasets are available at https://github.com/EricZhangSCUT/SPIN-CGNN.

## Abstract

Recent advances in deep learning have significantly improved the ability to infer protein sequences directly from protein structures for the fix-backbone design. The methods have evolved from the early use of multi-layer perceptrons to convolutional neural networks, transformers, and graph neural networks (GNN). However, the conventional approach of constructing K-nearest-neighbors (KNN) graph for GNN has limited the utilization of edge information, which plays a critical role in network performance. Here we introduced SPIN-CGNN based on protein contact maps for nearest neighbors. Together with auxiliary edge updates and selective kernels, we found that SPIN-CGNN provided a comparable performance in refolding ability by AlphaFold2 to the current state-of-the-art techniques but a significant improvement over them in term of sequence recovery, perplexity, deviation from amino-acid compositions of native sequences, conservation of hydrophobic positions, and low complexity regions, according to the test by unseen structures, "hallucinated" structures and diffusion models. Results suggest that low complexity regions in the sequences designed by deep learning, for generated structures in particular, remain to be improved, when compared to the native sequences.

## Author summary

We proposed SPIN-CGNN, a deep learning-based method for fixed backbone protein design. Our studies showed that introducing contact map-based graph construction, second-order edge updates, and selective kernels cumulatively improved the performance over existing methods in native and generated structures according to multiple measures, including amino-acid compositions, amino-acid substitutions, low complexity regions, and conservations of hydrophobic/hydrophilic positions for specific evaluation of fix backbone protein design methods.

**Funding:** This work was supported by National Key Research and Development Program of China [NO.2021YFF1200400] (XZ, HY, JZ, YZ) and Major Program of Shenzhen Bay Laboratory [S201101001] (XZ, HY, JZ, YZ). The funders had no role in study design, data collection and analysis, decision to publish, or preparation of the manuscript.

**Competing interests:** The authors have declared that no competing interests exist.

## 1. Introduction

De novo protein design is considered as an inverse problem of de novo protein structure prediction, that is, to find a sequence that would fold into a given structure, instead of predicting its structure for a given sequence. Both problems have been long dominated by energy-based approaches: energy-guided fragment reassembly in the case of protein structure prediction [1] and energy-guided sequence design in the case of protein design [2]. The progress for solving both problems (poor accuracy for de novo structure prediction and low success rate for designed sequences, respectively), however, were hampered by the lack of an accurate energy function to describe the solvent-mediated interactions between amino acid residues of proteins [2,3].

The energy functions for protein design were typically modified from protein folding studies that can be categorized as molecular mechanics force fields (e.g. EGAD [4]), statistical energy functions (e.g. ABACUS [5,6]), and mixed statistical, empirical, and physical force fields (e.g. RosettaDesign [7]). More recently, we developed a new protein design technique called OSCAR-design [8] that is based on a purely mathematical scoring function. This scoring function employed series expansion in distance and orientation dependence with mixing coefficients optimized for sequence recovery, sidechain modeling, and loop selection. The optimization effort leads to an average recovery of wild-type sequences at ~40%, similar to that achieved by the state-of-the-art technique RosettaDesign 3.12 with mixed physical and statistical energy terms, suggesting the bottleneck of an energy-based method.

To avoid an energy function, we developed the first direct prediction of sequences from structures by a simple artificial neural network called SPIN [9] (Sequence Profiles by Integrated Neural networks). By combining local-fragment-derived sequence profiles and nonlocal-energy functions, SPIN achieved a sequence recovery of 30% among 50 test proteins (denoted as TS50). SPIN2 [10] employed a deep three-layer neural network and additional structural features to improve SPIN, achieving a higher recovery of 34% on TS50. At the meantime, Qi et al. [11] employed a combination of three neural networks to predict amino-acid type of a center residue from structure fragments constructed with k-nearest neighbors (KNN) residues. With a preset k of 15, this method achieved 34% recovery in the 5-fold cross validation on a dataset constructed on PDB with 30% sequence identity cutoff. These early methods based on MLP (Multi-Layer Perceptrons), which is a basic architecture of artificial neural networks, provided a proof of concept for deep learning-based protein design given a fixed backbone structure.

Rapid advances in deep learning techniques enable the breakthrough of AlphaFold2 [12] and RoseTTAFold [13] in protein structure prediction by avoiding the need for an energy function through the end-to-end learning [14,15]. During the same period, there is a rapid employment of deep learning in protein design by convolutional neural networks (CNN) such as SPROF [16], DenseCPD [17], and ProDCoNN [18], graph neural networks (GNN) such as GraphTrans [19], GCA [20], GVP [21], AlphaDesign [22], ESM-IF [23], ProteinMPNN [24], PiFold [25], and LM-DESIGN [26], and transformer such as ABACUS-R [27] and ProDesign-LE [28]. These AI-driven backbone-based design substantially improved sequence recovery from 30–34% in MLP based techniques to ~50–55% by PiFold [25] or LM-DESIGN [26]. Moreover, the methods have developed from fixed backbone design, flexible design based on energy-based structure generation (SCUBA [29]) and sequence-based structure prediction [30,31] to structure and sequence generators [32–34].

This work focuses on the use of GNN for further improving AI-based fixed-backbone design as it appears to improve over MLP and CNN-based models in the latest development [35]. GraphTrans [19] represented protein backbone structures as a graph, in which residues

were represented by node features containing distance and orientation between sequential adjacent $C_\alpha$ atoms and backbone dihedral angles, while inter-residue distances and orientations were encoded as edge features. With an encoder-decoder model constructed with graph attention layers, node features were updated to predict the sequences by sequential autoregressive decoding. GraphTrans achieved 35.82% recovery after training and testing on a dataset set constructed on CATH 4.2 database (denoted as CATH4.2). GCA [20] (Global Context Aware generative protein design) improved GraphTrans by appending a global module after the local module (the graph attention layer). It improved the recovery on CATH4.2 to 37.64%. GVP [21] (Geometric Vector Perceptron) decoupled vector and scale information in graph features and proposed a network module to update geometrically sensitive representations. By simply replacing MLP layers employed in GraphTrans with GVP layers, the recovery on CATH4.2 increased to 39.47%. ProteinMPNN [24] improved GraphTrans with three modifications: replacing edge features with interatomic distances between all five atoms (including a virtual $C_\beta$ atom) on backbones, updating edge features in GNN, and replacing the sequential decoding order with random decoding order in autoregressive decoding. These modifications further improved sequence recovery on CATH4.2 to 45.96%. PiFold [25] introduced virtual atoms determined by backbone position and learnable parameters. Besides, the autoregressive decoding was replaced by one-shot decoding. PiFold achieved 51.66% recovery on CATH4.2 with orders of magnitude efficiency improvement.

The above methods can be further improved by using additional training data, scaling model sizes, and integration with large pretrained models. For example, ESM-IF [23] stack a scaled GVP model with a large transformer model and trained with over 1.2 million of structures predicted by AlphaFold2. The model containing over 142 million parameters significantly improved the recovery of the GVP model in CATH 4.2 dataset from 42.2% to 51.3%. LM-DESIGN [34] used a large protein language model, which was pretrained with over 50 million protein sequences, as a decoder to sampled protein sequences with an encoder from GVP, ProteinMPNN, and PiFold. With about 650 million of additional pretrained parameters, this method brought over 5% improvement to these methods. For the PiFold model reimplemented in this work, we found an increase from 50.22% to 55.65% for the sequence recovery.

These GNN-based methods [19–26], however, utilized k-nearest neighbors (KNN) to construct a graph for feature initialization and local information passing. The KNN graph construction significantly reduces the computational cost from passing information between all node pairs in a graph, since k is much smaller than the length of a sequence in most cases. Moreover, a proper setting of k is expected to prevent the modules for local information passing from overfitting the non-local information. Typically, k is set as 30 because many studies [19,24] suggested it sufficient for local information passing in GNN-based, fixed-backbone protein design. However, the local structures defined by a fixed number of neighbors might not be capable of handling dense local structures and sparse local structures at the same time.

In this study, we proposed SPIN-CGNN, a deep graph neural network-based method for the fixed backbone design, in which a protein structure graph is constructed with a distance-based contact map. This contact map-based graph construction (CGraph) enables GNN to handle a varied number of neighbors within a preset distance cutoff. In addition, we introduced information of symmetric and second order edges to update edge features. The symmetric edge information enabled information sharing inside an edge pair that connects two nodes. The information on second-order edges is expected to capture high-order interactions between two nodes from their shared neighbors. We found that this SPIN-CGNN achieved 54.81% for sequence recovery. This was achieved by employing a small model of 5.58 million parameters in the absence of pretrained models. Moreover, we further evaluated the method according to amino-acid substitution matrix, sequence complexity, and the deviation of the

query structure to the structure predicted by AlphaFold2. These performance measures further support the improvement of SPIN-CGNN over or comparable to existing state-of-the-art techniques.

## 2. Methods

Fig 1 shows the overall workflow of SPIN-CGNN. The backbone structure in SPIN-CGNN is first represented as a graph with a distance-based contact graph representation. It takes in the coordinates of backbone atoms including $N$, $C_\alpha$, $C$, $O$, and transforms them as node features for residues and edge features for inter-residues relationship. Moreover, the connections in the graph including edges and second-order edges were recorded for the computation of neural networks. Then, the represented graph will be inputted into the SPIN-CGNN blocks to iteratively extract structural features. An MLP will be applied on the updated node features to predict the probability of 20 amino acids for each position. Finally, protein sequences can be sampled based on the predicted probabilities.

### 2.1. Datasets

Many GNN-based methods have employed the CATH 4.2 dataset from Ingraham's [19] to assess their performance. This dataset was constructed by: a) collecting all chains with no more than 500 residues from the CATH 4.2 at 40% sequence identity cutoff; b) randomly split all collected chains into 80%/10%/10% subsets for training, validation, and test; c) removed the entries from these subsets to ensure that there was no overlap in CATH topology (also known as fold) classification between the subsets. Overall, the dataset consisted of a total of 18024, 608, and 1120 structure-sequence pairs in the training, validation, and test subsets from 950, 100, and 150 structural folds, respectively. Thus, there were overlaps in structural folds within each subset.

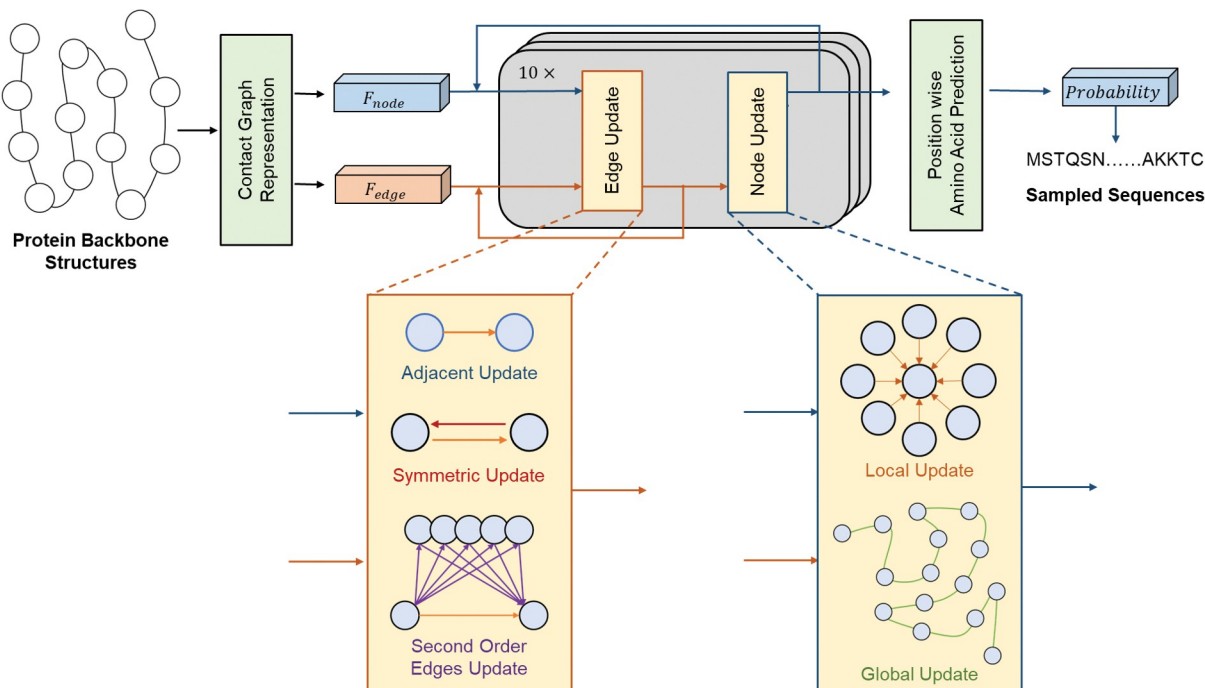

**Fig 1. The overall workflow of SPIN-CGNN, which employed a contact-based graph with improved edge and node updates.**

To avoid performance bias toward specific structural folds within the CATH4.2 test set, we calculated the TM-scores [36] between all test structures and found some structure pairs with high levels of similarity, as illustrated in S1 Fig. Thus, we created a non-structural redundant version of the test set by iteratively removing entries that had the most structural similar entries (TM-score > 0.4 [36]) in the test set until no structurally similar entry pairs remained. Ultimately, the resulting CATH4.2-StructNR193 test set contained 193 structure-sequence pairs, after removing 927 out of the original 1120 entries.

To generate another fully independent test set, we performed the following: a) gathering all PDB structures released after CATH4.2 (April 9[th], 2019); b) extracting chains with no less than 30 residues; c) calculating the TM-score between each chain and all existing entries from the new test set, the CATH4.2 test set, the CATH4.2 validation set, and the CATH4.2 training set; and d) retaining the chain if the maximum TM-score with all existing entries is no more than 0.4. Finally, we obtained a structural non-redundant test set with 156 entries, which we named as PDB-StructNR156.

Recently, several methods enabled the generation of near-native structures for protein design, including hallucination and diffusion models. We aimed to further evaluate the performance of fixed backbone protein design methods using generated structures. For hallucination methods, we utilized the 'Hallucination129' test set that includes 129 hallucinated structures [31]. For diffusion models, we constructed a new 'Diffusion100' test set by: a) generated 1000 structures with SE(3) diffusion model [37]; b) test the refoldability with the process from the paper of SE(3)-diffusion model. Specifically, we designed 8 sequences for each generated structure with ProteinMPNN, predicted structures for each sequence with ESMFold [38], calculated the median refold RMSD of 8 designed sequences; c) picked the 100 generated structures with the smallest median refold RMSD.

## 2.2. Graph representation

**Contact Map-based Graph Construction:** We defined the neighbors in a graph by contacts between the virtual $C_\beta$ atoms with a distance cutoff. The coordinate of virtual $C_\beta$ atom of each residue is calculated by

$$\mathbf{r}_{C_{\beta i}} = -0.58273431*u_i + 0.56802827*v_i - 0.54067466(u_i \times v_i) + \mathbf{r}_{C_{\alpha i}},$$

where $u_i$ is the directional vector from atom $C_{\alpha_i}$ to atom $N_i (u_i = \mathbf{r}_{C_{\alpha i}} - \mathbf{r}_{N_i})$, $v_i$ is the directional vector from atom $C_i$ to atom $C_{\alpha i}$ ($v_i = \mathbf{r}_{C_i} - \mathbf{r}_{C_{\alpha i}}$). Notably, we removed all self-loops in the contact graph (a residue is not a neighbor of itself). In the contact graph, a residue can receive information from different number of neighbors, depending on the local density around the residue. The edges in a contact graph are symmetric: whenever residue $i$ is a neighbor of residue $j$, residue $j$ would be a neighbor of residue $i$.

We constructed the node features and edge features as in PiFold [25]. In addition to conventional features widely employed such as distance and direction between backbone atoms, the PiFold features further incorporated bond lengths, bond angles, and learnable virtual atoms. A series of ablation studies conducted to discern the individual contributions of these components revealed a slight enhancement in the recovery performance of PiFold in fixed backbone design.

**Node Features**: For the node features of residue $i$, we defined a local coordinate system $Q_i = [b_i, n_i, b_i \times n_i]$ to construct rotation-invariant features, where $b_i = \frac{u_i - v_i}{\|u_i - v_i\|}$, and $n_i = \frac{u_i \times v_i}{\|u_i \times v_i\|}$. The unit directional vectors from $C_{\alpha_i}$ to $N_i$, $C_i$, $O_i$, and $C_{\beta_i}$ in the local coordinate system were collected as node directional features. Bond angles and torsion angles of continuous $N_i$, $C_{\alpha_i}$,

and $C_i$ were encoded as sine and cosine values as node angle features. Specifically, the bond angles included $C_{i-1}-N_i-C_{\alpha_i}$, $N_i-C_{\alpha_i}-C_i$, and $C_{\alpha_i}-C_i-N_{i+1}$, where $a-b-c$ denotes the angle between $\overrightarrow{ba}$ and $\overrightarrow{bc}$. The torsion angles included rotational angles around $\overrightarrow{C_{i-1}N_i}$, $\overrightarrow{N_iC_{\alpha_i}}$, $\overrightarrow{C_{\alpha_i}C_i}$ (i.e., $\omega$, $\phi$, $\psi$, respectively). The distances from $C_{\alpha_i}$ to $N_i$, $C_i$, $O_i$, and the virtual $C_{\beta_i}$ were encoded with the Gaussian radial basis function (RBF) from ProteinMPNN [24] as node distance features. Finally, the initial node features of residue $i$ were constructed by concatenating all unit vector, angle, and distance features.

**Edge Features**: The edge features of residue $j$ to residue $i$ also contain unit vector, angle, and distance features. All interatomic unit directional vectors from five atoms ($C_\alpha$, $C$, $N$, $O$, and the virtual $C_\beta$) of residue $j$ to those of residue $i$ were calculated and rotated with the local coordinate system $Q_i$, in total of 25 edge unit vector features. The interatomic distances between atoms of residue $i$ to atoms of residue $j$, including five main chain atoms ($C_\alpha$, $C$, $N$, and $O$ plus the virtual $C_\beta$) and three virtual atoms determined by learnable parameters, were collected and encoded with an RBF as 64 edge distance features in total. The coordinate of three virtual atoms $\{\mathbf{r}_{V_i^1}, \mathbf{r}_{V_i^2}, \mathbf{r}_{V_i^3}\}$ of residue $i$ can be calculated as

$$\begin{cases} \mathbf{r}_{V_i^n} = x_n u_i + y_n v_i + z_n(u_i \times v_i) + \mathbf{r}_{C_{\alpha_i}}, \\ x_n^2 + y_n^2 + z_n^2 = 1 \end{cases},$$

where $u_i = \mathbf{r}_{C_{\alpha_i}} - \mathbf{r}_{N_i}$, $v_i = \mathbf{r}_{C_i} - \mathbf{r}_{C_{\alpha_i}}$, and $(x_n, y_n, z_n)$ is a set of learnable parameters of the $n^{th}$ virtual atoms. These virtual atoms were employed for capturing complementary information with real atoms. The number of virtual atoms was set as 3 in this study, according to PiFold [25].

The rotation from $Q_j$ to $Q_i$ was encoded with the quaternion function as the edge angle features. Additionally, the sequential relative distance (the difference in sequence positions, $i-j$) from residue $j$ to residue $i$ was encoded, with a positional encoding function [39], and append into the edge features. Here, we set the dimension of positional encoding function and RBF as 16. Thus, the total dimensions of node and edge features are 96 and 1119, respectively. Note that any residues with any missing coordinates of atoms $C_\alpha$, $C$, $N$, and $O$ will be masked.

## 2.3. Network architecture

The graph neural network in SPIN-CGNN was built by stacking 10 Contact Graph Neural Network (CGNN) blocks to fit the message passing in contact map-based graphs. In a CGNN block, edge features were updated first (S2 Fig) and these updated edge features were then utilized to update node features (S3 Fig).

**2.3.1. Edge update.** Updating edge features has been proved to be useful for improving model performance in many related works including ProteinMPNN [24] and PiFold [25]. If we denote the node feature of node $i$ in layer $l$ as $h_i^l$ and the edge feature from $h_i^l$ to $h_j^l$ as $e_{ij}$, the edge update is performed by simply aggregating the information from adjacent nodes and edge itself:

$$\hat{e}_{ij}^{l+1} = MLP(h_i^l \parallel e_{ij}^l \parallel h_j^l),$$

where $\hat{e}_{ij}^{l+1}$ is the edge update by a MLP module, and $\parallel$ represents a concatenate operation.

We further enriched edge updates by edge symmetry. We defined $e_{ij}$ as the symmetric edge of $e_{ji}$ in a graph. To ensure that the information passing inside symmetric edge pairs is symmetric, each edge is concatenated with its symmetric edge to produce the symmetric edge

information $\hat{e}_{sym_{ij}}^{l+1}$ by:

$$\hat{e}_{sym_{ij}}^{l+1} = MLP(\hat{e}_{ij}^{l+1} \parallel \hat{e}_{ji}^{l+1}).$$

In addition to edge symmetry, we introduced the second-order edges (SOE) by considering the edges associated with their shared neighbors. More specifically, given $\mathcal{N}_i$ representing the neighbor nodes for node $i$, the shared neighbors of node $i$ and node $j$ can be represented as $\mathcal{N}_i \cap \mathcal{N}_j$. For a shared neighbor node $n \in (\mathcal{N}_i \cap \mathcal{N}_j)$, we defined the combination of $e_{in}$ and $e_{nj}$ as the SOE from node $i$ to node $j$ though node $n$. The edge update information from the SOE can be captured with a MLP module from the concatenated feature of $\hat{e}_{in}^{l+1}$, $\hat{e}_{nj}^{l+1}$, and $\hat{e}_{ij}^{l+1}$. By taking the average of all second-order edges of $e_{ij}$, we can calculate the SOE information $\hat{e}_{soe_{ij}}^{l+1}$ as:

$$\hat{e}_{soe_{ij}}^{l+1} = \frac{\sum_{n\in(\mathcal{N}_i\cap\mathcal{N}_j)}MLP(\hat{e}_{in}^{l+1} \parallel \hat{e}_{nj}^{l+1} \parallel \hat{e}_{ij}^{l+1})}{|\mathcal{N}_i \cap \mathcal{N}_j|},$$

where $|\mathcal{N}_i \cap \mathcal{N}_j|$ is the number of nodes in $\mathcal{N}_i \cap \mathcal{N}_j$. A second MLP module is applied to $\hat{e}_{ij}^{l+1}$ to specifically extract adjacent information $\hat{e}_{2_{ij}}^{l+1}$ from the basic edge update information $\hat{e}_{ij}^{l+1}$:

$$\hat{e}_{2_{ij}}^{l+1} = MLP(\hat{e}_{ij}^{l+1}).$$

The above updates were merged by a selective kernel module to produce the final edge update (S4A Fig), similar to the selective kernel convolution from SK-Net [40]:

$$\hat{e}_{graph_{ij}}^{l+1} = SK(\{\hat{e}_{sym_{ij}}^{l+1}, \hat{e}_{soe_{ij}}^{l+1}, \hat{e}_{2_{ij}}^{l+1}\}),$$

where $SK$ represents selective kernel, and $\hat{e}_{graph_{ij}}^{l+1}$ is the merged edge update information from the graph. To prevent overfitting, the edge information would be updated with dropout, residual connection, and layer normalization:

$$e'^{l+1}_{ij} = LayerNorm(e_{ij}^l + Dropout(\hat{e}_{graph_{ij}}^{l+1})).$$

Adding a position-wise feedforward module after attention module has been found to significantly improve the network performance [39]. Therefore, we performed the final update with such typical operation:

$$e_{ij}^{l+1} = LayerNorm(e'^{l+1}_{ij} + Dropout(FFN(e'^{l+1}_{ij}))),$$

where *FFN* represent the position-wise feedforward module.

**2.3.2. Node update.** Node updates in CGNN blocks were performed by integrating local information from neighboring nodes and global information from the whole graph. More specifically, we extract local information by aggregating information from neighboring nodes of the center node with a typical graph-attention module, in which the attention scores were calculated by:

$$\begin{cases} q_i^l = MLP(h_i^l) \\ k_{ji}^l = MLP(e_{ji}^l) \\ a_{ji}^l = \dfrac{\exp(q_i^l \cdot k_{ji}^l)}{\sum_{n\in\mathcal{N}_i}\exp(q_i^l \cdot k_{ni}^l)} \end{cases},$$

where $q_i^l$ is the query features of node $i$, $k_{ji}^l$ is the key features of $e_{ji}^l$, and $a_{ji}^l$ is the attention score of $e_{ji}^l$ for the local information of node $i$.

We further calculated the value features $v_{ji}^l$ which were concatenated from edge feature $e_{ji}^l$, and its adjacent node features $h_j^l$ and $h_i^l$. The summation of neighboring attention-scaled value features yields the local information $\hat{h}_{local_i}^{l+1}$:

$$\begin{cases} v_{ji}^l = MLP(h_j^l \parallel e_{ji}^l \parallel h_i^l) \\ \hat{h}_{local_i}^{l+1} = \sum_{j \in \mathcal{N}_i} a_{ji}^l v_{ji}^l \end{cases}.$$

In addition to the local updates, we also used global updates to account for nonlocal interactions. If we define $\mathcal{G}$ as all nodes in a protein structure graph, we can calculate the global context $G^l$ by summing scaled value features from all nodes, in which both value feature and attention were calculated from the node feature itself as below:

$$\begin{cases} a_i^l = \dfrac{\exp(MLP(h_i^l))}{\sum_{i \in \mathcal{G}} \exp(MLP(h_i^l))} \\ v_i^l = MLP(h_i^l) \\ G^l = \sum_{i \in \mathcal{G}} a_i^l v_i^l \end{cases}.$$

The global update $\hat{h}_{global_i}^{l+1}$ was then calculated from the concatenated features of node feature and global context, with an MLP:

$$\hat{h}_{global_i}^{l+1} = MLP(h_i^l \parallel G^l).$$

Similarly, a selective kernel module (S4B Fig) was used to merge global with local update information:

$$\hat{h}_{graph_i}^{l+1} = SK(\{\hat{h}_{local_i}^{l+1}, \hat{h}_{global_i}^{l+1}\}).$$

Same as edge update, the node features were updated with the graph update information:

$$h\prime_i^{l+1} = LayerNorm(h_i^l + \hat{h}_{graph_i}^{l+1}),$$

followed by a *FFN* update:

$$h_i^{l+1} = LayerNorm(h\prime_i^{l+1} + Dropout(FFN(h\prime_i^{l+1})))$$

to reduce the possibility of overfitting.

**2.3.3. Selective kernel.** The selective kernel from SKNet [40] (Selective Kernel Networks) was designed for and has been widely used to merge multi-scale features captured by convolutional kernels with different kernel sizes. In SPIN-CGNN, it was employed to adaptively merge a set of features from different update modules.

Given a feature set $\mathcal{F}$ containing $n$ features, a selective kernel simply summarizes all features and squeeze the dimension with an MLP:

$$f_{squeeze} = MLP(\sum_{i=1}^{n} f_i).$$

A set of MLPs were employed for the excitation from $f_{squeeze}$ to the dimension-wise weight of each feature, and normalized by *Softmax* function:

$$w_i = MLP_i(f_{squeeze}),$$

$$a_i = \frac{w_i}{\sum_{i=1}^{n} \exp(w_i)}.$$

Finally, the merged feature $f_{merge}$ is calculated by summarizing all features that are dimension-wisely scaled by the attention:

$$f_{merge} = \sum_{i=1}^{n} a_i f_i.$$

## 2.4. Training

The dimensions of edge and node features were set as 128 for all layers. All models were trained by minimizing the cross entropy between output logits and native sequences for 100 epochs with AdamW optimizer [41]. The learning rate was adjusted according to OneCycle learning rate schedule [42] with a max learning rate of 0.004. We set the drop probability as 0.1 for all dropout operators. Training data were randomly grouped with a maximum batch size of 4096 residues. We employed mixed precision to accelerate the training speed and reduce the GPU memory occupation in all experiments. All other settings followed the default setting of PyTorch.1.13 [43].

## 2.5. Performance measure

The most widely used criteria for evaluating the methods for fixed backbone design are perplexity and recovery. The perplexity on test set $\mathcal{D}$ is calculated by exponentiated categorical cross-entropy loss per residue:

$$Perplexity(\mathcal{D}) = \exp\left(-\frac{\sum_{S^N \in \mathcal{D}} \sum_{i=1}^{N} S_i^N \log S_i^{N'}}{\sum_{S^N \in \mathcal{D}} N}\right),$$

where $S^N$ is a sequence with $N$ residues from the test set $\mathcal{D}$. $S_i^N$ is the $i$-th native residue and $S_i^{N'}$ is the corresponding predicted probability from the model. Perplexity is a measure that accounts for the certainty around the native amino acid residues. Lower perplexity values indicate smaller deviation from native residue types.

Recovery, measuring the ability of the model to reconstruct the native sequence of a protein, is calculated by the percentage of the identity of designed sequences to native sequences:

$$Recovery(S^N, S^{N'}) = \frac{1}{N} \sum_{i=1}^{N} 1\left[S_i^N = argmax(S_i^{N'})\right].$$

This measure, however, only compared the top ranked prediction and does not reflect fluctuation around the top ranked prediction.

We further examined the frequencies of each amino-acid-residue type given by native sequences and designed sequences (amino acid compositions). The similarity between native frequencies and predicted frequencies can be measured by the relative deviations:

$$Rel.Dev.\left(X_i, X_i'\right) = \frac{|X_i - X_i'|}{X_i},$$

where $X$ is native frequencies and $X'$ is the predicted frequencies.

It is known that surface residues are more difficult to recover, we calculated the fraction of surface residues for each target protein. The residue-wise SASA (Solvent Accessible Surface Area) was obtained by BioPython [44]. The SASA for each residue is divided by the maximum allowed solvent accessibility (MaxASA) [45] of the residue type to yield the relative accessible surface area (RSA). Finally, we classified the residues with RSA smaller than 0.2 as core residues, and the others as surface residues as in OSCAR-design [8].

However, the above criteria are based on native sequences. Many sequences can fold into the same structure. Some sequences can fail to fold into a target structure despite high sequence identity to the native sequence because a few mutations may well destabilize the structure. Thus, we also examined low complexity regions, which is the subsequences that normally lead to intrinsically disordered regions and the inability to fold into the target structure. We detected these subsequences using the SEG algorithm [46]. SEG identifies approximate segments of low complexity using a sliding window in the first pass and optimized these segments in the second pass. The optimized segments with the information measure lower than a given threshold will be marked as an LCR. Specifically, we run the SEG algorithm with the default setting from NCBI C++ toolkit [47], which is 12 for sliding window size and 2.2 for low-information cutoff. The fractions of low complexity regions were calculated as a criterion for analysis, denoted as LCR. Additionally, we measured the frequencies of amino acid substitutions in the designed sequences from the native sequences by calculation the BLOSUM score and the Pearson correlation coefficient of the confusion matrix with BLOSUM62[48] as the reference of native amino acids substitution. The BLOSUM score is calculated as the summation of BLOSUM62 values of the native amino acid, weighted by the predicted probability. It should be mentioned here that we did not calculate BLOSUM score for OSCAR-design and RosettaFixBB because it does not yield the probability of amino acid residues in one design as in AI-based methods. Although in principle one could perform 100 designs for each protein to obtain the probability, computational requirement is prohibitive for our available computing resource when applying to the large dataset we are employing here for test. The calculation of confusion matrix followed ESM-IF [23], in which the substitution scores between native sequences and designed sequences were calculated by using the same log odds ratio formula as: in the BLOSUM62 substitution matrix. For two amino acid types $x$ and $y$, the substitution score is:

$$s(x, y) = \log \left( \frac{p(x, y)}{q(x)q(y)} \right),$$

where $p(x, y)$ is the jointly likelihood that native amino acid $x$ is substituted by predicted amino acid $y$, $q(x)$ is the frequencies of amino acid $x$ in the native distribution, and $q(y)$ is the frequencies of amino acid $y$ in the predicted distribution.

Finally, we performed a test to evaluated the performance of methods by measuring the structure deviations between the target structures and the AlphaFold2-predicted structures for designed sequences. Notably, we run AlphaFold2 with default setting while excluding PDB templates to avoid the influence of the template searched by high sequence recovery. Three criteria were employed including TM-score, RMSD (Root-Mean-Square Deviation), GDT-TS (Global Distance Test–Total Score) [49] on the superposition $C_\alpha$ coordinate. Specifically, the distance cutoff used in GDT-TS is 1, 2, 4, and 8 Å, same as that in CASP [49].

## 2.6. Method comparison

We employed four methods for comparison including two energy-based methods Rosetta-FixBB and OSCAR-design, and two GNN-based method ProteinMPNN and PiFold. For the

energy-based method RosettaFixBB and OSCAR-design, we designed sequences with the default setting. For deep learning-based methods ProteinMPNN and PiFold, we reimplemented them with the source code and the training setting from their paper. Specifically, we reimplemented ProteinMPNN model with its source code and training setting: negative-log likelihood loss, transformer learning rate schedule, batch size 6000 residues, training epochs 100, Adam optimizer, 30 residue neighbors, no coordinate noise. The median recovery of the reimplemented model tested on CATH4.2 test set is 46.15%, which is consistent to the reported recovery of ProteinMPNN reimplemented model (45.96%) [25]. We also reimplemented PiFold with its source code and training setting: negative-log likelihood loss, OneCycle learning rate schedule, a batch size of 4096 residues, training epochs of 100, Adam optimizer, 30 residue neighbors, and 3 virtual atoms. The median recovery of the reimplemented model tested on CATH4.2 test set is 51.55%, which is also consistent to the reported recovery (51.66%) [25].

## 3. Results

### 3.1. Impact of graph constructions

We examined the impact of using the contact maps at different cutoff distances and compared them against the K-nearest-neighbor graph (k = 30, KNN-30) employing the CATH4.2--StructNR193 and the PDB-StructNR156 test sets. Performance was evaluated using perplexity and median recovery. To make a fair comparison, Table 1 compares KNN-30 to CGNN all at without employing CGNN edge information (named as Model 1). The results indicated that KNN-30 has a better performance than CGraph-8 (Contact graph at 8Å distance cutoff) for both test sets. However, increasing distance cutoff (from 8Å to 10Å, and then 12 Å) improves over KNN-30 in perplexity and sequence recovery. At 12 Å cutoff, there is ~1% increase in median sequence recovery, and 2–3% reduction of perplexity from KNN-30 to CGraph-12. We note that even at 12Å cutoff, the average number of neighbors (25 or 29) is still smaller than 30 employed in KNN-30. We fixed the cutoff at 12Å for all subsequent analysis because further increasing the cutoff will only lead to minor improvement at the expense of higher computational requirement.

**Table 1. Contact-based versus K-nearest neighbors in the absence of edge information for all methods (Model 1 for SPIN-CGNN) according to perplexity and median sequence recovery for two test datasets (CATH4.2-StructNR193 and PDB-StructNR156).**

| Graph Construction [a] | #Neighbors [b] | Perplexity ↓[c] | Median Recovery (%) ↑[c] |
|---|---|---|---|
| CATH4.2-StructNR193 | | | |
| KNN-30 | 30 | 4.65 ± 0.04 | 51.05 ± 0.32 |
| CGraph-8 (Model 1) | 9 | 4.70 ± 0.02 | 50.39 ± 0.41 |
| CGraph-10 (Model 1) | 16 | 4.55 ± 0.04 | 51.55 ± 0.37 |
| CGraph-12 (Model 1) | 25 | **4.54 ± 0.03** | **52.07 ± 0.39** |
| PDB-StructNR156 | | | |
| KNN-30 | 30 | 3.75 ± 0.02 | 56.45 ± 0.21 |
| CGraph-8 (Model 1) | 10 | 3.84 ± 0.03 | 54.67 ± 0.44 |
| CGraph-10 (Model 1) | 17 | 3.65 ± 0.01 | 56.69 ± 0.13 |
| CGraph-12 (Model 1) | 29 | **3.64 ± 0.01** | **57.12 ± 0.24** |

[a] KNN: K-nearest neighbors, with K = 30; CGraph-8, 10, 12: the contact graphs at 8, 10, and 12Å cutoffs.

[b] The number of neighbors is fixed for KNN-30 but is an averaged number for CGraph-8, CGraph-10, and CGraph-12.

[c] The average of 5 parallel tests with the standard deviations.

**Table 2. Impact of CGNN edge updates (symmetric information and second-order edge (SOE) information) according to perplexity and median sequence recovery for two test datasets (CATH4.2-StructNR193 and PDB-StructNR156).**

| Model | Sym. | SOE | Perplexity ↓[a] | Median Recovery (%) ↑[a] |
|---|---|---|---|---|
| CATH4.2-StructNR193 | | | | |
| Model 1 | × | × | 4.54 ± 0.03 | 52.07 ± 0.39 |
| Model 2 | √ | × | 4.43 ± 0.02 | 52.55 ± 0.40 |
| Model 3 | × | √ | **4.36 ± 0.01** | **53.04 ± 0.31** |
| SPIN-CGNN | √ | √ | **4.36 ± 0.01** | 52.89 ± 0.64 |
| PDB-StructNR156 | | | | |
| Model 1 | × | × | 3.64 ± 0.01 | 57.12 ± 0.24 |
| Model 2 | √ | × | 3.53 ± 0.01 | 57.50 ± 0.48 |
| Model 3 | × | √ | 3.46 ± 0.02 | 58.45 ± 0.50 |
| SPIN-CGNN | √ | √ | **3.43 ± 0.02** | **58.54 ± 0.27** |

[a] The average of 5 parallel tests with the standard deviations.

## 3.2. Ablation test for CGNN edge updates

Table 2 examines the effect of second-order edge (SOE) and symmetric edge updates by constructing Model 2 and Model 3, respectively, as well as the SPIN-CGNN with both SOE and symmetric edge updates. Table 2 shows that without both edge updates Model 1 leads to the worst performance in perplexity and median sequence recovery. Removing SOE also led to statistically significant increase of perplexity and reduction of median recovery from SPIN-CGNN. Although symmetric edge updates do improve the CGNN model when SOE update is absent (from Model 1 to Model 2), it contributes little when the SOE update is performed, indicating that the information captured by symmetric edge updates may have been covered by SOE updates. The overall effect of introducing edge updates is ~1% increase in sequence recovery and 4–6% reduction in perplexity.

## 3.3. Ablation test for selective kernel (SK)

Table 3 examines the effect of the feature integrating module, selective kernels (SKs), compared to average pooling on features to be integrated. Specifically, we constructed three additional models: Model 4, where all SKs in both edge and node updates were replaced by average pooling, Model 5, where only SKs in edge updates were replaced, and Model 6, where SKs in only node updates were replaced. Table 3 indicates that Model 4 had the worst or second-to-the-worst perplexity and median sequence recovery. The cumulative improvement due to the use of SK is 3% reduction in perplexity and 1% increase in sequence recovery. We also evaluated the effect of SK with comparison to MLP modules (S1 Table) and observed similar improvement of using SK.

We noted that there is a large gap in the performance between CATH4.2-StructNR193 and PDB-StructNR156 test sets. We found that this is mainly because surface residues are less conserved and, thus, harder to recover in computational design and the PDB-StructNR156 test set has 6.5% less surface residues (54.5% versus 61.0%), and thus, with about 5% higher sequence recovery for designed sequences than the CATH4.2-StructNR193 test set. As shown in S5 Fig, there is an overall similarity in the dependence of recovery on the fraction of surface residues, indicating the robustness of SPIN-CGNN on unseen structures. Moreover, Deep-learning-based methods consistently outperform energy-based techniques (OSCAR-design and RosettaFixBB) in both core and surface regions.

**Table 3. Impact of the use of selective kernels in node update and edge update according to perplexity and median sequence recovery for two test datasets (CATH4.2-StructNR193 and PDB-StructNR156).**

| Model | Node SK | Edge SK | Perplexity ↓ [a] | Median Recovery (%) ↑ [a] |
|---|---|---|---|---|
| CATH4.2-StructNR193 | | | | |
| Model 4 | × | × | 4.48 ± 0.01 | 51.75 ± 0.86 |
| Model 5 | √ | × | 4.44 ± 0.01 | 52.47 ± 0.44 |
| Model 6 | × | √ | 4.39 ± 0.01 | 52.43 ± 0.34 |
| SPIN-CGNN | √ | √ | **4.36 ± 0.01** | **52.89 ± 0.64** |
| PDB-StructNR156 | | | | |
| Model 4 | × | × | 3.55 ± 0.02 | 57.40 ± 0.58 |
| Model 5 | √ | × | 3.56 ± 0.01 | 57.63 ± 0.18 |
| Model 6 | × | √ | 3.44 ± 0.02 | 57.71 ± 0.16 |
| SPIN-CGNN | √ | √ | **3.43 ± 0.02** | **58.54 ± 0.27** |

[a] The average of 5 parallel tests with the standard deviations.

### 3.4. Method comparison on the whole CATH4.2 test set

Table 4 compared SPIN-CGNN to a number of other methods for fixed-backbone protein design that employed the same CATH4.2 training, validation and test sets. This is based on the whole test set (rather than structurally non-redundant set) as we do not have the performance for all individual proteins for most methods. As we can see, SPIN-CGNN achieved the best performance in terms of both perplexity and recovery for the whole test set, as well as for two subsets of small and single-chain proteins with 3–4% improvement of recovery and 10–20% improvement in perplexity over the next best PiFold for those methods without a pretrained language model. Compared to LM-DESIGN, which employed the language model for enhancing the method PiFold, our method continues to improve over perplexity by 10% with a slightly lower sequence recovery (1.0%).

### 3.5. Method comparison on structural non-redundant test sets

To confirm that the above improvement by SPIN-CGNN over other methods was not due to biased structural redundancy, Table 5 compared the performance of SPIN-CGNN, OSCAR-design, ProteinMPNN, and PiFold on CATH4.2-StructNR193 and PDB-StructNR156 test sets. Here, we employed RosettaFixBB and OSCAR-design as examples of the energy-based techniques, PiFold and ProteinMPNN as examples of modern deep learning models. The results confirmed that SPIN-CGNN has the lowest perplexity (10–15% reduction from the second-best method PiFold, highest sequence recovery (3–4% increase from PiFold).

### 3.6. Method comparison on sequence compositions of amino acid residues

Obviously, fluctuation around native sequences (perplexity) and the recovery of native sequences are only one aspect to measure the quality of predicted sequences. The diversity of amino acid residues employed is another measure for designed sequences. A well-designed sequence should take the advantage of the diversity of amino acid residues. Fig 2 compares the frequency of each amino acid residue types employed in native sequences and in the sequences designed by SPIN-CGNN, RosettaFixBB, OSCAR-design, ProteinMPNN, and PiFold, for CATH4.2-StructNR193 (A) and PDB-StructNR156 (B) test sets. There is a large deviation of ProteinMPNN from native frequencies due to its over-employment of A, E, L, and V and under-employment of H, M, Q, R, and W, as shown in Fig 2. The imbalance of residue usages

**Table 4. Method comparison on the whole CATH4.2 test set according to perplexity and median native sequence recovery.**

| Methods | Perplexity ↓ | | | Median Recovery (%) ↑ | | |
|---|---|---|---|---|---|---|
| | Short[a] | Single-chain[b] | All | Short[a] | Single-chain[b] | All |
| Energy-based Protein Design | | | | | | |
| RosettaFixBB | - | - | - | 18.49 | 19.05 | 31.18 |
| OSCAR-design | - | - | - | 23.18 | 21.48 | 34.83 |
| AI-based Protein Design | | | | | | |
| StructGNN[c] | 8.29 | 8.74 | 6.40 | 29.44 | 28.26 | 35.91 |
| GraphTrans[c] | 8.39 | 8.83 | 6.63 | 28.14 | 28.46 | 35.82 |
| GCA[c] | 7.09 | 7.49 | 6.05 | 32.62 | 31.10 | 37.64 |
| GVP[c] | 7.23 | 7.84 | 5.36 | 30.60 | 28.95 | 39.47 |
| AlphaDesign[c] | 7.32 | 7.63 | 6.30 | 34.16 | 32.66 | 41.31 |
| ProteinMPNN | | | | | | |
| - implemented by Gao et al[c] | 6.21 | 6.68 | 4.61 | 36.35 | 34.43 | 45.96 |
| - implemented in this work | 6.28 | 6.71 | 4.68 | 33.58 | 31.65 | 46.15 |
| PiFold | | | | | | |
| - implemented by Gao et al[c] | 6.04 | 6.31 | 4.55 | 39.84 | 38.35 | 51.66 |
| - implemented in this work | 5.93 | 6.15 | 4.59 | 38.10 | 37.61 | 51.55 |
| LM-DESIGN + PiFold[d] | 6.77 | 6.46 | 4.52 | 37.88 | 42.47 | **55.65** |
| SPIN-CGNN | **5.01** | **5.02** | **4.05** | **44.71** | **43.37** | 54.81 |

[a] Short: the sub test set for those targets with less than 100 amino acids.

[b] Single-chain: the sub test set for those targets from single-chain structures.

[c] Results adapted from Ref. [25].

[d] Results adapted from Ref. [26].

by ProteinMPNN led to the highest median relative deviation of 0.357 (0.306), compared to 0.141 (0.169) by RosettaFixBB, 0.145 (0.110) by PiFold, 0.099 (0.063) by SPIN-CGNN, and 0.078 (0.067) by OSCAR-design for the CATH4.2-StructNR193 (PDB-StructNR156) dataset (Table 5). Thus, SPIN-CGNN (and OSCAR-design) has much more natural sequence compositions than PiFold and ProteinMPNN (SPIN-CGNN is 32 or 43% better than PiFold, depending on the dataset).

### 3.7. Substitutions between amino acids

The phenomenon of amino acid substitutions offers the possibility of different sequences to attain the same target protein structure. This is due to the fact that some positions in the protein structure permit the interchange of amino acids without affecting structural stability. We obtained amino acid substitutions in fixed backbone protein design methods by computing the position-wise confusion matrix between the predicted and native amino acids. Such confusion matrix can be used to compare to BLOSUM62 matrix, that describes the likelihood of amino acid replacements in native sequences. As shown in Fig 3, we can see the confusion matrix of SPIN-CGNN presented a similar pattern to the reference BLOSUM62 matrix: most positive substitutions in BLOSUM62 matrix are also positives values in the confusion matrix of SPIN-CGNN. The Pearson correlation coefficient of SPIN-CGNN on CATH4.2-StructNR193 test set was calculated to be 0.899, compared to 0.841 by RosettaFixBB, 0.884 by OSCAR-Design (S6 Fig), 0.839 by ProteinMPNN (S7 Fig), and 0.890 by PiFold (S8 Fig). We also obtained the correlation coefficients of these methods on PDB-StructNR156 test set. Similarly, the correlation coefficient given by SPIN-CGNN (0.869) is higher than that of

**Table 5. Comparison of sequences designed by SPIN-CGNN, RosettaFixBB, OSCAR-design, ProteinMPNN, and PiFold on CATH4.2-StructNR193 and PDB-StructNR156 test sets according to perplexity, median sequence recovery, median relative deviation of the frequency of amino-acid residue types, the median relative BLOSUM score, the fraction of low complexity regions, conservation of hydrophobic and hydrophilic sequence positions, the mean steric clash count of refolded structures, and the difference between refolded and target structures in term of RMSD, GDT-TS and TM-score.**

| Methods | Perplexity ↓ | Median Recovery (%) ↑ | AA Compositions (Median Rel. Dev.) ↓ | AA Substitutions (Median Relative BLOSUM score)↑ | LCR (%) ↓ | Hydrophobic Conservation (%) ↑ | AlphaFold2 Prediction Test | | | |
|---|---|---|---|---|---|---|---|---|---|---|
| | | | | | | | Median RMSD (Å) ↓ | Median GDT-TS ↑ | Median TM-score ↑ | Mean Clash Count ↓ |
| CATH4.2-StructNR193 | | | | | | | | | | |
| Native (Reference) | - | - | - | - | 4.12 | - | 1.85 | 83.82 | 0.863 | 0.021 |
| RosettaFixBB | - | 29.29 | 0.141 | - | 10.14 | 68.16 | 5.01 | 52.91 | 0.759 | **0.053** |
| OSCAR-design | - | 33.33 | **0.078** | - | **8.35** | 71.22 | 3.17 | 63.95 | 0.819 | 0.088 |
| ProteinMPNN | 4.97 | 43.82 | 0.357 | 0.362 | 23.95 | 74.02 | 2.75 | 67.97 | 0.812 | 0.130 |
| PiFold | 4.88 | 50.00 | 0.145 | 0.394 | 13.31 | 80.99 | 2.41 | 75.88 | 0.847 | 0.321 |
| SPIN-CGNN | **4.36** | **52.50** | 0.099 | **0.442** | 11.22 | **82.50** | **2.09** | **79.13** | **0.857** | 0.078 |
| PDB-StructNR156 | | | | | | | | | | |
| Native (Reference) | - | - | - | - | 4.27 | - | 1.89 | 83.68 | 0.903 | 0.944 |
| RosettaFixBB | - | 34.78 | 0.169 | - | 8.30 | 70.40 | 2.58 | 72.09 | 0.869 | **0.879** |
| OSCAR-design | - | 39.41 | 0.067 | - | 7.80 | 74.17 | 2.62 | 75.79 | 0.890 | 1.246 |
| ProteinMPNN | 4.04 | 47.26 | 0.306 | 0.433 | 12.93 | 77.07 | 2.21 | 76.50 | 0.891 | 1.855 |
| PiFold | 4.00 | 54.81 | 0.110 | 0.459 | 6.56 | 82.56 | 1.88 | 81.17 | 0.899 | 1.779 |
| SPIN-CGNN | **3.42** | **58.88** | **0.063** | **0.517** | **5.49** | **85.44** | **1.75** | **85.65** | **0.913** | 1.101 |

RosettaFixBB (0.850), ProteinMPNN (0.853) and PiFold (0.863). Notably, the energy-based method OSCAR-design outperformed all deep learning-based methods with the highest coefficients of 0.888 for this test set. We also calculated the BLOSUM score, a summation of BLOSUM62 values weighted by the predicted probability, as a composite metric of perplexity and amino acid substitution. The BLOSUM score of the methods was further normalized by dividing it with the BLOSUM score of the native sequences, where the probabilities of residues were substituted with the one-hot encoded native sequence. As presented in Table 5, SPIN-CGNN outperformed ProteinMPNN and PiFold on both CATH4.2-StructNR193 and PDB-StructNR156 test sets with respect to the median relative BLOSUM score (0.442 / 0.517 for SPIN-CGNN, 0.362 / 0.433 for ProteinMPNN, and 0.394 / 0.459 for PiFold). These results highlight the stronger overall ability of SPIN-CGNN to capture evolution information than other deep learning techniques.

## 3.8. Sequence complexity

The presence and distribution of low-complexity regions (LCR) within protein sequences plays a crucial role in both their structural and functional properties, making it a vital aspect of protein design. Higher LCR fractions in designed sequences as compared to native sequences may result in protein's structural instability.

The fractions of LCR for native sequences are at 4.12% and 4.27% for the CATH4.2-StructNR193 and the PDB-StructNR156 test sets, respectively. All designed sequences had more LCRs as shown in Table 5. SPIN-CGNN has the lowest (5.5% for the PDB-StructNR156 test set) or the third lowest (11.2% for the CATH4.2-StructNR193, behind OSCAR-design and RosettaFixBB) fractions of LCRs. Compared to other deep learning techniques, SPIN-CGNN is 1%-2% improvement over PiFold. ProteinMPNN has the worst performance as expected because it over-employed small hydrophobic residues such as A, L, and V (Fig 2).

## 3.9. Hydrophobicity conservation

One important requirement for soluble proteins is that hydrophobic residues should be mostly buried inside the core whereas surface residues are dominated by hydrophilic residues to ensure solubility and prevent hydrophobic aggregation. We examined the conservation of hydrophobic and hydrophilic sequence positions of design sequences by defining hydrophobic (Ile, Leu, Met, Phe, Cys, Trp, Pro, Val, Ala and Gly) and hydrophilic (Ser, Thr, Asn, Gln, Asp, Glu, His, Arg, Lys and Tyr) residue positions according to the native sequence. As Table 5 shows that SPIN-CGNN has the highest conservation in hydrophobicity positions (~3% over the next best PiFold) for both non-redundant test sets.

## 3.10. Deviation of target structures from the structures predicted by AlphaFold2 based on designed sequences

To further evaluate whether the designed sequences would fold into target structures as expected, we employed AlphaFold2 [12] without using templates as a part of input to predict

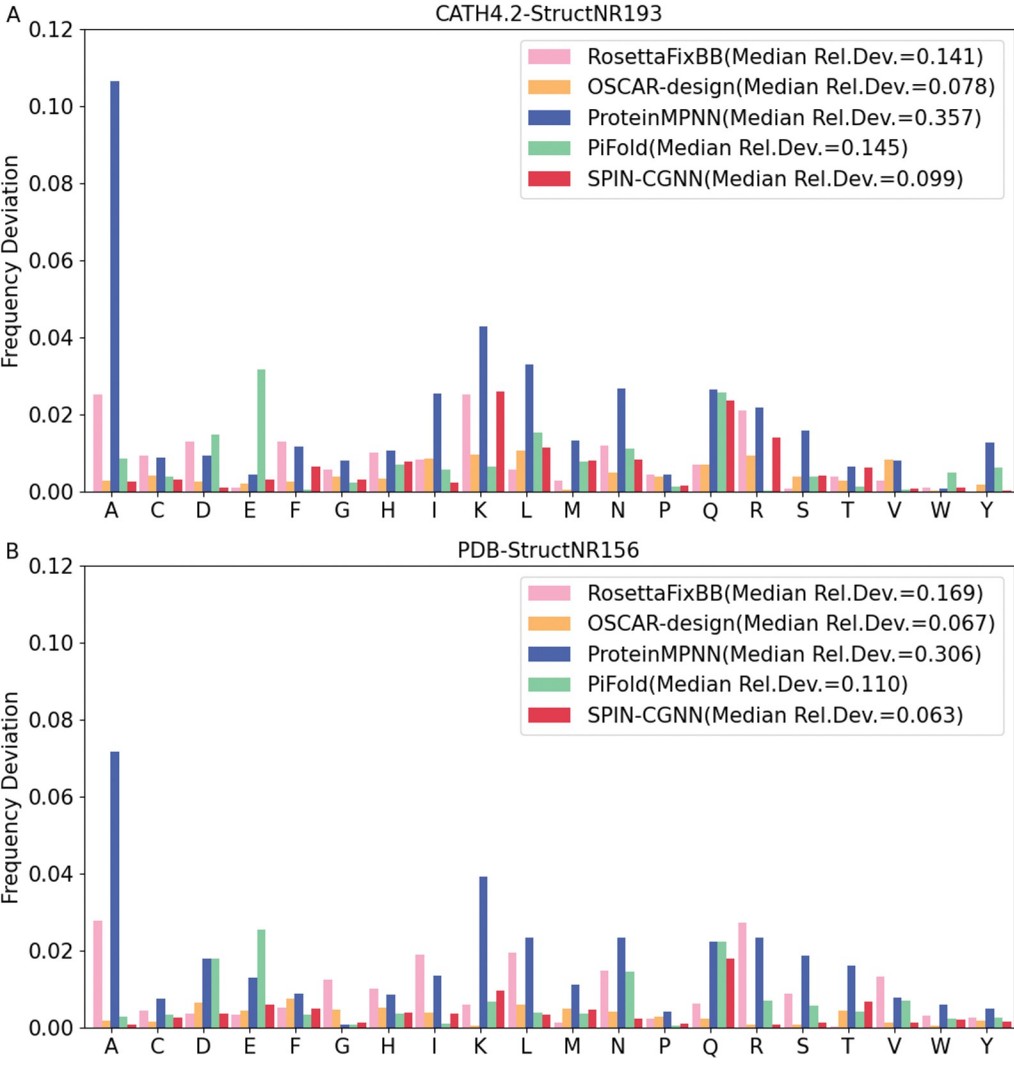

**Fig 2. Deviation of the frequency of an amino acid in designed sequences from that in the native sequences by RosettaFixBB, OSCAR-design, ProteinMPNN, PiFold and SPIN-CGNN.** (A) CATH4.2-StructNR193 and (B) PDB-StructNR156 test set.

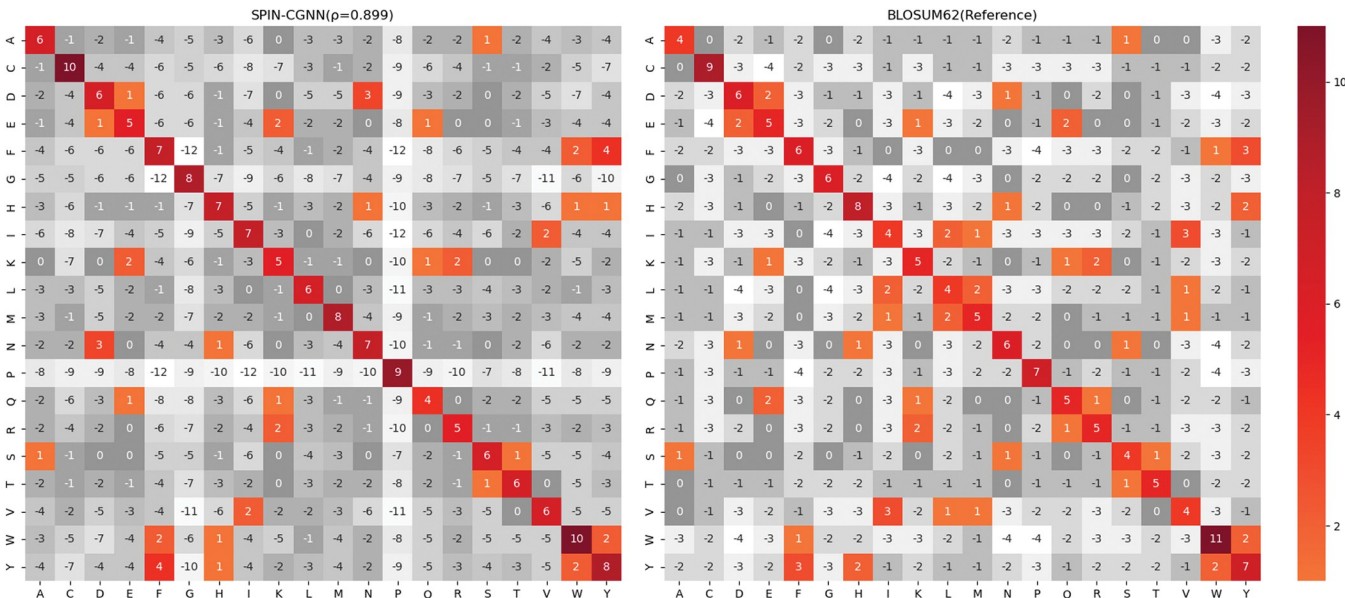

**Fig 3. Confusion matrix of SPIN-CGNN in comparison to the reference matrix BLOSUM62 on the CATH4.2-StructNR193 test set.** Positive values (colored) indicate substitutions between amino acids. $\rho$ denotes the Pearson correlation coefficient between confusion matrix of SPIN-CGNN and BLOSUM62.

the structures of designed sequences and measured the root-mean-square deviation (RMSD), global distance test-total score (GDT-TS), and TM-score between predicted structures and target structures. As shown in Table 5, the predicted structures of sequences designed by SPIN-CGNN achieved the smallest median RMSD of 2.09 Å, the greater median GDT-TS of 79.13, and the highest median TM-score of 0.857 on the CATH4.2-StructNR193 test set, comparing to that of RosettaFixBB (5.01 Å, 52.91, and 0.759, respectively), OSCAR-design (3.17 Å, 63.95, and 0.819, respectively), ProteinMPNN (2.75 Å, 67.97, and 0.812, respectively), and PiFold (2.41 Å, 75.88, and 0.847, respectively). As a reference, we also run AlphaFold2 on native sequences and measured the structure deviation (1.85 Å, 83.82, and 0.863, respectively). We displayed the refoldability of different methods as a distribution in Fig 4. PiFold and SPIN-CGNN have comparable performance (no statistically significant difference) in term of refoldability by AlphaFold2. SPIN-CGNN also have comparable performance to native sequences, in term of RMSD on CATH4.2-StructNR193. We noted that including templates in AlphaFold2 prediction only leads to small improvement in refolded structures as shown in S2 Table and S9 Fig.

Additionally, we count the steric clash within each AlphaFold2-predicted structures and compare SPIN-CGNN to other methods according to the mean clash count, as shown in Table 5. The mean clash count of SPIN-CGNN on CATH4.2-StructNR193 is 0.078, which is the second lowest comparing to OSCAR-design (0.088), ProteinMPNN (0.130) and PiFold (0.321), behind RosettaFixBB (0.053). Notably, the mean clash count of RosettaFixBB and ProteinMPNN could be under-estimated due to their over-employment of small residues such as Ala.

### 3.11. Performance on test sets of generated structures

To further evaluate SPIN-CGNN and compare with other methods, we expand our experiments with two generated structure test sets. The Hallucination129 test set is made of the structures generated by hallucination and, thus, each structure does not have a native sequence

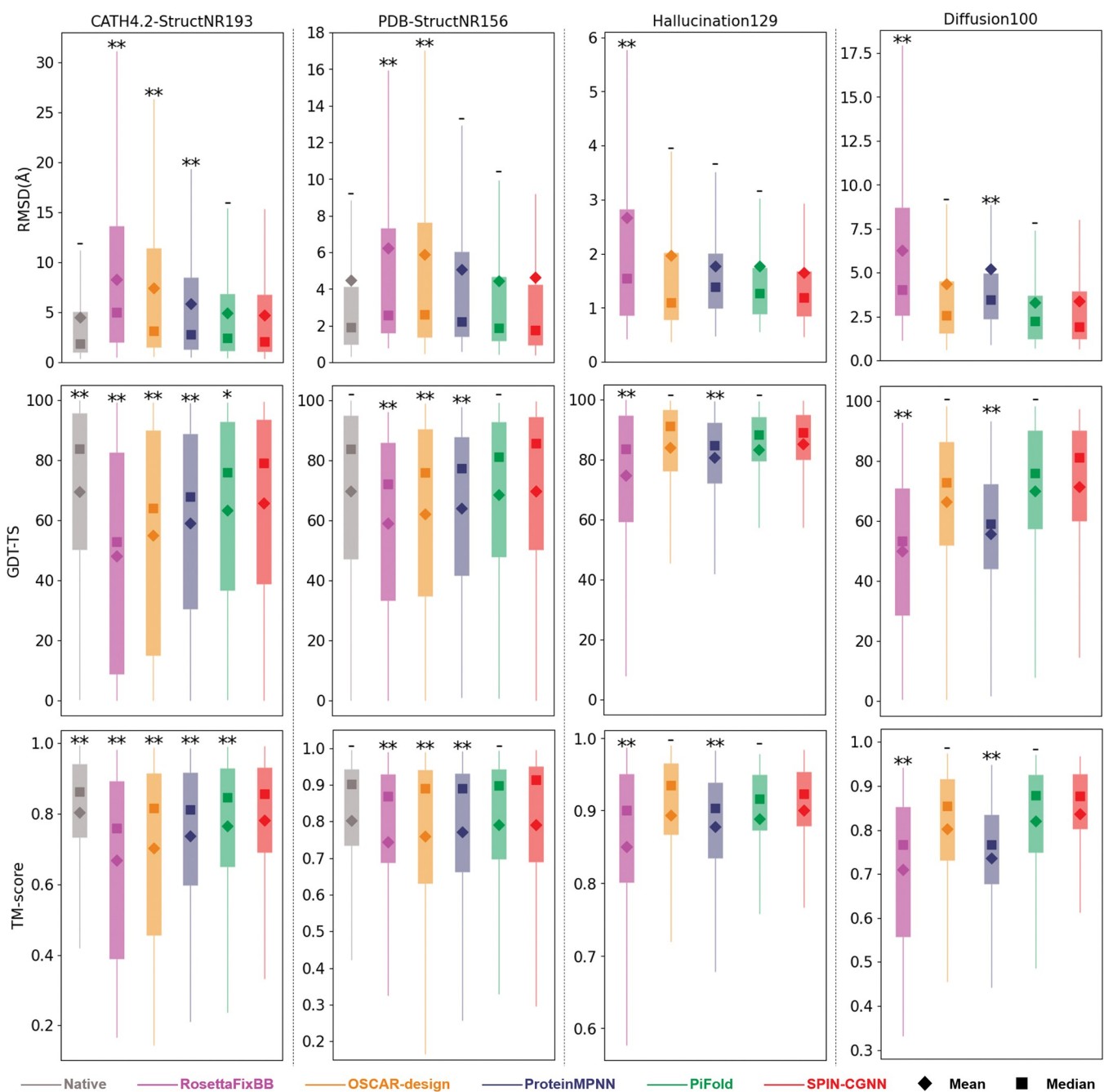

**Fig 4.** Deviations of the structures of designed sequences predicted by AlphaFold2 from their respective target structures on three separate test sets from left to right panels (CATH4.2-StructNR193, PDB-StructNR156, Hallucination129, and Diffusion100 test sets) evaluated according to RMSD (Å), GDT-TS, and TM-score (from top to bottom panels). The statistical significance of the difference of a given method to SPIN-CGNN was marked with '**' for highly statistically significant (p-value<0.01), '*' for statistically significant (0.01<p-value<0.05), and '-' for not statistically significant (p-value>0.05). Specific p-values are presented in S4 Table.

to calculate sequence recovery. The same is true for the Diffusion100 set. As shown in Table 6, OSCAR-design has the lowest fraction of LCR (11%), compared to 15% by SPIN-CGNN, 17% by PiFold, 20% by RosettaFixBB and 33% by ProteinMPNN. For the difference between target structures and AlphaFold2-predicted structures (Fig 4), only RosettaFixBB and ProteinMPNN

has highly significant worse performance on GDT-TS and TM-score, and RMSD from PiFold, SPIN-CGNN, and OSCAR-design. The deviations of refolded from native structures given by PiFold, SPIN-CGNN, and OSCAR-design are statistically similar to each other. Adding Gaussian noise of 0.02 Å standard deviation to coordinates did not lead to further improvement of deep learning techniques on artificially generated structures, unlike a previous report [24] (S3 Table).

On the Diffusion100 test set, which is made of structures generated by a diffusion model, the refoldabilities of all methods are worse than that on the Hallucination129 test set, which may be possibly due to the worse designability of this diffusion model. Compared to the other methods, SPIN-CGNN have better refoldability than RosettaFixBB and ProteinMPNN, while comparable to OSCAR-design and PiFold (with statistically insignificant difference). It's worth mentioning that LCRs of deep learning methods are even higher (23.16% for SPIN-CGNN, 32.73% for PiFold, and 56.32% for ProteinMPNN) than that on the Hallucination129 test set. By comparison, the LCRs of sequences designed by OSCAR-design on the Diffusion100 test set (5.01%) is significantly lower than all the other methods, demonstrating its robustness on sequence complexity. Additionally, we calculated AA composition deviations for methods on both generated structures test sets, with the average native AA composition from the CATH4.2-StructNR193 and PDB-StructNR156 test set as a reference (S10 Fig). The AA composition deviations of all methods are significantly higher than that on native structures. There are two possible reasons: a) design methods are not robust for unseen structures; b) the generated structures are biased toward most popular amino acid residues due to fixed backbone conformations (See more in the discussion section).

### 3.12. Case study

To further understand how SPIN-CGNN improved fixed backbone protein design with contact graph (CGraph), we presented a case in Fig 5, where the structure of NADP-reducing hydrogenase subunit HndA (PDB 2AUV Chain A) was employed for protein design. SPIN-CGNN outperformed PiFold with a higher recovery (30.59% vs. 28.24%) and a much smaller refold RMSD (5.66 Å vs. 14.02 Å). The CGraph12 graph construction method captured

**Table 6. Comparison of sequences designed by SPIN-CGNN, ProDesign-LE, RosettaFixBB, OSCAR-design, ProDesign-LE, ProteinMPNN, and PiFold on the Hallucination129 and Diffusion100 test sets according to the fraction of Low-Complexity Regions (LCR), the mean steric clash count of refolded structures, and the difference between refolded and target structures in term of RMSD, GDT-TS and TM-score.**

| Methods | LCR (%) ↓ | AA Compositions (Median Rel. Dev.) ↓[a] | AlphaFold2 Prediction Test | | | |
|---|---|---|---|---|---|---|
| | | | Median RMSD (Å) ↓ | Median GDT-TS ↑ | Median TM-score ↑ | Mean Clash Count ↓ |
| Hallucination129 | | | | | | |
| ProDesign-LE[28] | - | - | - | - | 0.83 (Mean) | - |
| RosettaFixBB | 19.94 | 0.412 | 1.55 | 83.50 | 0.901 | 0.016 |
| OSCAR-design | 11.00 | 0.390 | **1.10** | **91.25** | **0.935** | **0.000** |
| ProteinMPNN | 32.96 | 0.565 | 1.38 | 84.75 | 0.903 | 0.023 |
| PiFold | 17.19 | **0.289** | 1.26 | 88.25 | 0.916 | 0.078 |
| SPIN-CGNN | 14.91 | 0.387 | 1.19 | 89.00 | 0.923 | 0.039 |
| Diffusion100 | | | | | | |
| RosettaFixBB | 14.12 | 0.406 | 4.04 | 53.25 | 0.768 | **0.000** |
| OSCAR-design | **5.01** | **0.322** | 2.54 | 72.88 | 0.855 | 0.040 |
| ProteinMPNN | 56.32 | 0.518 | 3.47 | 59.00 | 0.767 | 0.080 |
| PiFold | 32.73 | 0.386 | 2.26 | 76.00 | **0.879** | 0.190 |
| SPIN-CGNN | 23.16 | 0.362 | **1.91** | **81.13** | 0.878 | 0.070 |

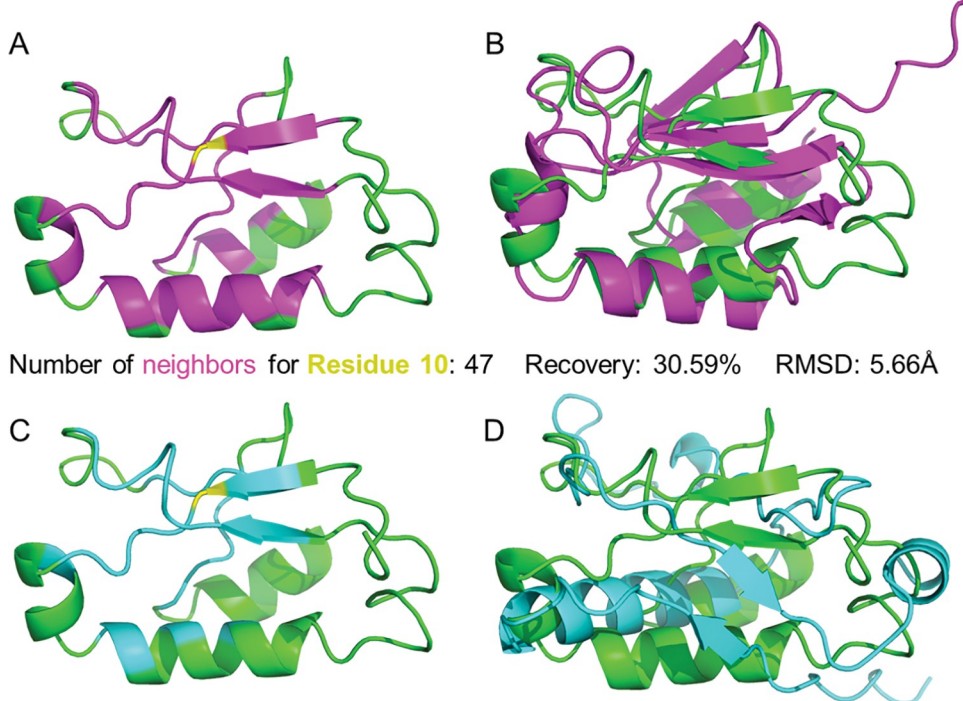

**Fig 5. An illustrative example to highlight the dense local contacts accounted by SPIN-CGNN for improving fixed backbone design.** (A) The neighbors for residue 10 (yellow) in PDB 2AUV chain A, which has the greatest number of neighbors determined by CGraph12 (magenta). (B) The AlphaFold2-predicted structure of sequence designed by SPIN-CGNN (magenta), aligned on the native PDB structure (green). (C) The neighbors determined by KNN-30 (cyan) for the same protein. (D) The corresponding AlphaFold2-predicted structure of sequence designed by PiFold (cyan).

more information from the compact core, as shown in Fig 5A with 47 neighboring residues for residue 10, compared to a fixed number of 30, when KNN-30 were employed.

## 4. Discussion

We proposed SPIN-CGNN, a deep learning-based method for fixed backbone protein design. Our approach incorporates several key elements, including contact-map graph construction, second-order edge updates, and selective kernels. Ablation studies reveal that the power of SPIN-CGNN lies in the cumulative improvement resulting from these multiple changes. Comparing our method with the most recently developed deep learning-based approaches, ProteinMPNN and PiFold, we found that SPIN-CGNN consistently outperformed them across various metrics including perplexity, sequence recovery, amino-acid composition, amino-acid substitution, low complexity regions, and conservations of hydrophobic/hydrophilic positions. Refolding of designed sequences by AlphaFold2 indicates that the structures produced by SPIN-CGNN are comparable to those by PiFold but significantly closer to target structures than ProteinMPNN.

Interestingly, a recently developed energy-based technique, OSCAR-design, produced comparable performance to PiFold and SPIN-CGNN for structures refolded by AlphaFold2 for Hallucination129 and Diffusion test sets. Depending on the datasets, OSCAR-design can have the closest amino-acid composition and low complexity region to the native composition

compared to ProteinMPNN, PiFold and SPIN-CGNN, indicating that there is something that deep learning techniques can be improved further.

When we detected the structural redundancy within the CATH4.2 test set, we also found structure pairs with high structure similarity (TM-score>0.4) between all three CATH4.2 sub-sets (training, validation, and test). To reduce the potential of overfitting, we removed all structures in the training set that have TM-score >0.4 with any structures in the validation and test sets. This led to a much smaller training set of 9311 structures, compared to 18024 proteins in the original set. This new training set, however, led to a poorer performance for those unseen structures. For example, the median TM-score refolded by AlphaFold2 was reduced from 0.924 trained by the whole training set to 0.823 trained by the new training set for SPIN-CGNN for the PDB StructNR156. The similar behavior was observed for PiFold and ProteinMPNN. The worse performance by using TM-score <0.4 is not only because it is too strict for removing many nonredundant folds but also because TM-score is just a global metric and the larger set can keep more abundant local structural information which are very useful for training. In this case, it will be difficult to separate the contributions from these two different effects. Thus, we employed the whole training set as it improves the generalizability over the smaller training set.

We would like to emphasize the importance of using different measures to computationally assess the designed sequences. This is because native sequence recovery and the deviation from the native sequence (perplexity) only reflect one aspect of designed sequences. High sequence identity does not assure foldability as a few mutations are often found sufficient to disrupt structural stability. Moreover, too many hydrophobic residues on the protein surface will lead to insoluble and aggregated proteins and low complexity in sequences often leads to intrinsically disordered regions. In addition, the refoldability by AlphaFold2 is not that sensitive to the sequences given by different methods because designed sequences are now highly similar to the wild type sequences and, thus, naturally occurring homologous sequences were employed as a part of the input for AlphaFold2 (Tables 5 and 6). Here we showed that the fraction of low complexity regions remains much higher than native sequences and energy-based techniques, indicating the room for further improvement.

It is well known that backbone bond angles and torsion angles may contain information biased toward certain amino acid residues. For example, the $N-C_\alpha-C$ angle is within [121, 126] for histidine and within [117, 122] for leucine. Unlike other residues, Gly does not any forbidden regions in the $\phi-\psi$ torsion angle space. This leads to the question if removing these angles will lead to a significant change in method performance. To address this question, we trained a SPIN-CGNN model with inter-atomic distances as the edge features only. We found that the recovery of the distance-only model on native structures drops only slightly from 52.5% to 51.21% on the CATH4.2-StructNR193 test set and from 58.8% to 57.36% on the PDB-StructNR156 test set. Thus, the angles only have a small impact on residue-type recovery.

Two sets of generated model protein structures were also employed for testing various protein design techniques. They are Hallucination and Diffusion sets. Table 6 shows that although most methods can refold designed sequences to the corresponding generated structures well, both physical and AI-based methods have elevated factions of low-complexity regions and larger deviation of amino acid compositions from natural proteins. This may be because "generated" structures, particularly from diffusion models, do not explicitly consider specific side-chains. As a result, the geometrical characters of main chains and the physical space between mainchain atoms in these designed structures maybe more favorable for most popular residue types (as shown in Fig 2). These structures would be prone to produce sequences with low complexity, regardless of physical or deep-learning based methods.

It is noted that the number of parameters employed by SPIN-CGNN is 5.58 million, comparable to 4.13 million by PiFold. It has a similar inference time as PiFold. For a 500-residue protein, the inference time is 0.09 second by SPIN-CGNN, compared to 0.03 second by PiFold, 0.83 by ProteinMPNN.

## Supporting information

**S1 Fig. High structural similarity pairs (TM score > 0.98) within the CATH4.2 test set.** The alignments of structure pairs were presented between two test structures in green and cyan.
(TIF)

**S2 Fig. Edge updating in the CGNN block.**
(TIF)

**S3 Fig. Node updating in the CGNN block.**
(TIF)

**S4 Fig.** Selective kernels for edge update (A) and node update (B) in the CGNN block.
(TIF)

**S5 Fig.** The median sequence recovery of protein targets as a function of the fraction of surface residues on CATH4.2-StructNR193 (A) and PDB-Struct156 (B) test sets given by SPIN-CGNN, in comparison with a number of other methods as labeled. Nearly identical dependence on fraction of surface residues by SPIN-CGNN for two different test sets indicates the robustness of the methods for different datasets.
(TIF)

**S6 Fig. Confusion matrix given by OSCAR-design.**
(TIF)

**S7 Fig. Confusion matrix given by ProteinMPNN.**
(TIF)

**S8 Fig. Confusion matrix given by PiFold.**
(TIF)

**S9 Fig.** Deviations of the structures of designed sequences predicted by AlphaFold2 with PDB templates from their respective target structures on four separate test sets from left to right panels (CATH4.2-StructNR193, PDB-StructNR156, Hallucination129, and Diffusion100 test set) evaluated according to RMSD (Å), GDT-TS, and TM-score (from top to bottom panels). The statistical significance of the difference of a given method to SPIN-CGNN was marked with '**' for highly statistically significant (p-value<0.01), '*' for statistically significant (0.01<p-value<0.05), and '-' for not statistically significant (p-value>0.05).
(TIF)

**S10 Fig. Deviation of the frequency of an amino acid in designed sequences from that in the native sequences by RosettaFixBB, OSCAR-design, ProteinMPNN, PiFold and SPIN-CGNN.** (A) Hallucination129 and (B) Diffusion100 test set.
(TIF)

**S1 Table. Impact of the use of selective kernels in node update and edge update according to perplexity and median sequence recovery for two test datasets (CATH4.2-StructNR193 and PDB-StructNR156).** Two-layer MLP modules were employed to substitute SK modules

in models without selective kernels (no-SK models).
(DOCX)

**S2 Table. Refoldability tested by AlphaFold2 with PDB templates.**
(DOCX)

**S3 Table. Adding Gaussian noises to the structural coordinates slightly improved the performance of deep-learning methods (SPIN-CGNN, ProteinMPNN, and PiFold) on the Hallucination129 test set, according to the fraction of Low-Complexity Regions (LCR) and the difference between refolded and target structures in term of RMSD, GDT-TS and TM-score, except an increase of low complexity regions for SPIN-CGNN and PiFold but a reduction of LCR for ProteinMPNN.**
(DOCX)

**S4 Table. Statistical significance between a given method to SPIN-CGNN for the structural difference between target structures and AlphaFold2-predicted structures for designed sequences according to Root Mean Square Deviation (RMSD), Global Distance Test-Total Score (GDT-TS) and TM-Score for three test sets.** '**' denoted p-value < 0.01, '*' denoted 0.01 < p-value < 0.05, and '-' denoted P-value > 0.05.
(DOCX)

## Acknowledgments

We thank Professor Dongbo Bu for his suggestion of the method for the BLOSUM score calculation. The work was done by using the supercomputing facility of the Shenzhen Bay Laboratory.

## Author Contributions

**Conceptualization:** Xing Zhang, Jian Zhan, Yaoqi Zhou.

**Data curation:** Xing Zhang, Yaoqi Zhou.

**Formal analysis:** Xing Zhang.

**Funding acquisition:** Jian Zhan, Yaoqi Zhou.

**Investigation:** Xing Zhang, Hongmei Yin, Yaoqi Zhou.

**Methodology:** Xing Zhang, Yaoqi Zhou.

**Project administration:** Jian Zhan, Yaoqi Zhou.

**Resources:** Fei Ling, Jian Zhan, Yaoqi Zhou.

**Software:** Xing Zhang, Hongmei Yin.

**Supervision:** Fei Ling, Jian Zhan, Yaoqi Zhou.

**Validation:** Xing Zhang, Hongmei Yin, Fei Ling.

**Visualization:** Xing Zhang.

**Writing – original draft:** Xing Zhang, Yaoqi Zhou.

**Writing – review & editing:** Xing Zhang, Hongmei Yin, Fei Ling, Jian Zhan, Yaoqi Zhou.

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
