## [Decision Letter · Decision Letter 0]

7 Sep 2023

Dear Dr Zhou,

Thank you very much for submitting your manuscript "SPIN-CGNN: Improved fixed backbone protein design with contact map-based graph construction and contact graph neural network" for consideration at PLOS Computational Biology.

As with all papers reviewed by the journal, your manuscript was reviewed by members of the editorial board and by several independent reviewers. In light of the reviews (below this email), we would like to invite the resubmission of a significantly-revised version that takes into account the reviewers' comments.

We cannot make any decision about publication until we have seen the revised manuscript and your response to the reviewers' comments. Your revised manuscript will be sent to reviewers for further evaluation.

Sincerely,

Yang Zhang

Guest Editor

PLOS Computational Biology

Nir Ben-Tal

Section Editor

PLOS Computational Biology

Reviewer's Responses to Questions

**Comments to the Authors:**

Reviewer #1: The study by Zhang et al. presents a new deep learning method for fixed backbone protein sequence design, SPIN-CGNN. The overall architecture is similar to the latest graph-based neural network methods such as ProteinMPNN and PiFold. The major new advancements in the paper are adaptive construction of graph edges through specific distance thresholds, the use of selective kernel networks for node and edge updates, as well as the introduction of symmetric and second-order edge information. The work is nicely presented and should be suitable for publication after revisions are made.

My specific comments may be found below.

1. I appreciate the effort the authors put into retraining ProteinMPNN and PiFold; however, the authors should compare the performance of these programs based on the publicly released, pretrained models by each respective lab. It is not the author’s responsibility to improve the pipeline or training procedure for the control methods. Moreover, others could argue that the results are worse due to the retraining and not because of methodological improvements by SPIN-CGNN. Thus, the results of the default models should be provided whenever possible. If redundancy between the training and test sets is an issue, the authors may examine the performance on a set of non-redundant targets.

2. The authors state on lines 506-507 in reference to Table 4, “we do not have the performance for all individual proteins for most methods.” Did the authors take the results on the test set from the papers directly? How do we know that the reported results are accurate? The authors should download each program and run them locally on their test set if possible to verify the performance of each method. This should be done for all results presented in the manuscript.

3. I would suggest that the authors add in a comparison with RosettaFixBB and/or RosettaFlexibleBB, which are probably two of the most widely-used energy function-based methods for protein sequence design.

4. What is the formula for calculating the LCR in Tables 5 and 6?

5. In Table 5, what is the sequence recovery rate in buried vs solvent exposed regions of the protein. Are the deep learning methods very good at recovering buried regions but not at recovering exposed regions? Do the physical energy-based methods outperform the deep learning methods on the exposed regions?

6. In Table 5, what is the BLOSUM score for OSCAR-design. Currently, the BLOSUM score is calculated based on the probability distribution similar to the perplexity, making it impossible to calculate this value for OSCAR-design. However, the authors may additionally calculate the BLOSUM score based on the actual designed sequences similar to the sequence recovery rate. Do the physics-based methods have more favorable BLOSUM scores than the deep learning methods? And if so, what does this mean?

7. In Table 6, what is the sequence recovery rate and AA composition deviation for each of the methods on the Hallucination129 test set?

8. In Tables 5 and 6, how was AlphaFold2 run? Were just the single designed sequences used or did the authors use MSAs? Also, were templates with >30% sequence identity to the queries excluded. If not, the TM-scores may be better for the deep learning methods as the native structures themselves could be identified as templates given the high sequence identity between the native and designed sequences.

9. The results in Tables 5 and 6 are very interesting. For example, the median sequence recovery rate for SPIN-CGNN is nearly 60% higher than the physical energy function-based method, OSCAR-design; however, OSCAR-design produces sequences with AA compositions closer to the native distributions. Does this mean that the OSCAR-design sequences are more physically realistic? Additionally, why are the results for the deep learning methods significantly better than the physical energy function-based methods on the native structures but worse on the designed structures? Are SPIN-CGNN and the other deep learning-based methods very good at recapitulating high-confidence, conserved regions in the protein structures but fail to recapitulate less conserved regions? What are the physical characteristics of the designs produced by the deep learning methods? Do they introduce steric clashes or physically unrealistic residues? I believe a closer examination of the physical quality and stability of the designs is warranted. The authors could run MD simulations to verify the stability of the designed sequences and provide a closer examination of the physical quality of the designs by examining such things as steric clashes and MolProbity scores, among others.

Reviewer #2: Zhang et al. presented SPIN-CGNN as a graph neural network-based deep learning approach for de novo protein sequence on fixed protein backbones. Compared to the conventional GNN methods based on kNN graphs (where k is typically a value fixed to 30), SPIN-CGNN constructs graphs based on the distance-based protein contact maps, thus allowing an adaptive “k” for different local structures of the backbone. They also introduced symmetric and second order edge updates in their GNN. The approach, though lacking a wet-lab validation, is well benchmarked in silico with a variety of metrics, including native sequence recovery, perplexity, amino acid composition, and amino acid substitution, to just name a few. SPIN-CGNN achieved SOTA performance on almost all metrics, with the exception that the relatively high ratio of low-complexity regions in designer sequences compared to native sequences. I recommend publishing this method paper after a few minor comments are addressed.

1. In lines 200-202, the u_i and v_i should be defined in line 192.

2. Lines 423-424, “ProteinMPNN reimplemented model” is somewhat confusing. I found that ProteinMPP was also reimplemented in ref. 25 and the rate of 45.96% was from the reproduced version therein. Thus, I suggest the authors change “ProteinMPNN reimplemented model” to “ProteinMPNN implemented by Gao et al.”

3. In Table 2, “4.36±0.01” in the “Model 3” row should also be highlighed.

4. In Table 4, SPIN-CGNN outperformed “LM-DESIGN + PiFold” all categories of perplexity test and the “Short” and “Single-chain” categories of Median recovery (~7% and ~1%, respectively) but was ~1% worse in “All” of Median recovery. Could the authors compare the performance of SPIN-CGNN and “LM-DESIGN + PiFold” on the targets that are not categorized as “Short” or “Single-chain”, so that readers can understand how much worse SPIN-CGNN on these targets?

5. The concept of “low complexity region” should be clearly defined in the manuscript. My understanding is the “loop” or “coil” region of a protein, but it should be clearly defined.

6. The performance varies a lot between two test sets in Table 5. Instead of just listing the results there, I think it would be more interesting to explain these differences in greater detail. Could the authors add the LCR (%) for native sequences in this table as a reference? Nevertheless, is LCR (%) a good metric for assessing the design quality as they showed the AlphaFold2 prediction test? I also so because in CATH4.2-StructNR193, the LCR by OSCAR-design was the lowest, but its corresponding AlphaFold2 prediction test was the worst (in terms of RMSD and GDT-TS)? If we trust the global structure prediction metric, it seems that a lower LCR does not necessarily lead to a higher refolding ability.

7. In lines 693-694, “This new training set, however, led to a poorer performance for those unseen structures.” And lines 698-699 “we employed the whole training set as it improves the generalizability over the smaller training set.” Could the authors explain more reason on the poorer performance on the nonredundant, smaller training set. Is this because the “TM-score<0.4” is too strict, removing many nonredundant folds? Or is because TM-score is just a global metric and the larger set can keep more abundant local structural information which are very useful for training, and TM-score<0.4 will eliminate many useful local information?

8. The authors have tons of data, but they did not dig them very deeply, nor the Discussion section was discussed in depth.

Reviewer #3: In the manuscript, the authors present a GNN-based approach, called SPIN-CGNN, to protein sequence design. The motivation of the study is to circumvent the drawbacks of constructing GNN using k-nearest neighbors with $k$ fixed beforehand. The SPIN-CGNN approach incorporates the node/edge feature settings proposed by PiFold, the message passing mechanism proposed by ProteinMPNN and GraphTrans, and two mechanisms for edge updating (second-order edge and symmetric edge) proposed by the authors themselves. The authors conducted extensive experiments (including recovery, perplexity, the predicted structures of the designed sequences, residue diversity, sequence complexity) to examine the performance of SPIN-CGNN.

The idea of the SPIN-CGNN is interesting; however, I still have several concerns.

Major concerns:

1. Presentation of the approach: In the manuscript, the authors described the node updating and edge updating only and did not mention the procedure for sequence generating, although they showed two blocks in Figure 1. The authors should describe the complete procedure of sequence design to make readers easy to follow.

2. Dominating factors: the approach is a mixture of a large variety of ingredients, including using virtual atoms and bond angle/length as node features, using inter-atomic vectors and distances as edge features, and using SK mechanism to merge multiple features. The mixture of multiple ingredients is useful; however, this also brings a question: Which ingredient is the dominating one that determines residue type at a position?

3. Robustness of the approach: the authors used bond angles as node features, which carry information of residue type, for example, the N-C-CA angle is within [121, 126] for histidine and within [117, 122] for leucine. This fact raises a question: is bond angle the key feature for determining residue type? If yes, does this mean that this approach relies on accurate structure by X-ray crystallography? Is this the reason that the performance of the approach is not that good on hallucinated proteins as on CATH proteins? I also suggest the authors to examine the performance of SPIN-CGNN on the structures generated by diffusion model.

4. Redundancy of node features: In addition, the authors used bond lengths as node features; however, the bond length is roughly the same for all residues. The effects of virtual atoms is also not clearly stated. Are these features really inforamtive for residue type determination?

5. Organization of the Result section: I appreciate the author’s viewpoints that, for protein sequence design, the quality of the predicted structure is much more important than sequence recovery and perplexity. The current form of the manuscript puts “impact of graph construction” and two ablation studies as the first three results. I suggest the authors to reorganize the Result section with the quality of the predicted structures (subsection 3.11) as the first subsection. I also suggest the authors to add a column into Table 1, 2, 3, 4 to show the quality the predicted structures.

6. Inconsistent figures: In both Table 2 and 3, the mean perplexity and recovery of SPIN-CGNN on CATH4.2 is 4.36 and 52.89, respectively. However, these figures are 4.05 and 54.81 in Table 4. Why? In addition, the head of Table 1, 2, 3 is “mean recovery” but the authors listed them in the form of standard variation.

7. The effect of SK: The SK technique calculates the sum of multiple features (Line 334) and then applies the multi-head mechanism on it. However, the simple addition of the features, say symmetric edge features, SOE and the extracted features (Line 281), might make one type of the features dominate the others. How about apply MLP on the concatenation of these features?

8. The effect of SOE: The message-passing mechanism aims to broadcast the information of the neighbors to the central residue even the neighbors are 1, 2, 3, … hops apart and thus naturally considers the information of secondary-order neighbors. The authors should perform case study to clearly show the effects of SOE updating.

9. Case study: I suggest the authors to provide case study to clearly state the most effective ingredients of the approach.

Minor concerns:

1. Grammar errors:

I suggest the authors to polish the manuscript to remove grammar errors, say:

Line 405: “we performed a test to evaluated ….”

2. Position of punctuations:

In Line 192, the sentence starts with a comma. Please put the comma symbol after the formula.

3. Inconsistent use of punctuations:

The authors use “a)” in Line 155 but use “(a)” in Line 173.

4. “Marked as TS50” => “denoted as TS50”

**Have the authors made all data and (if applicable) computational code underlying the findings in their manuscript fully available?**

Reviewer #1: Yes

Reviewer #2: Yes

Reviewer #3: Yes

PLOS authors have the option to publish the peer review history of their article (what does this mean?). If published, this will include your full peer review and any attached files.

Reviewer #1: No

Reviewer #2: No

Reviewer #3: No
---

## [Decision Letter · Decision Letter 1]

21 Nov 2023

Dear Dr Zhou,

Thank you very much for submitting your manuscript "SPIN-CGNN: Improved fixed backbone protein design with contact map-based graph construction and contact graph neural network" for consideration at PLOS Computational Biology. As with all papers reviewed by the journal, your manuscript was reviewed by members of the editorial board and by several independent reviewers. The reviewers appreciated the attention to an important topic. Based on the reviews, your manuscript may become acceptable, providing that you modify the manuscript according to the recommendations of the third reviewer. Meanwhile, please provide appropriate references to TM-score and GDT-score which have been extensively used in the manuscript but are not necessarily known to all readers. Meanwhile, necessary references are needed for justifying some of the specific score cutoffs (e.g., TM-score>0.4).

Sincerely,

Yang Zhang

Guest Editor

PLOS Computational Biology

Nir Ben-Tal

Section Editor

PLOS Computational Biology

Reviewer's Responses to Questions

**Comments to the Authors:**

Reviewer #1: The authors have appropriately address all my concerns and I believe the paper is acceptable for publication.

Reviewer #2: The authors have tried their best to address my questions and comments. I suggest accepting the manuscript in its current form.

Reviewer #3: I appreciate the authors’ efforts to address my concerns. However, I have some further concerns:

1. Dominating factors: the approach is a mixture of a large variety of ingredients and the authors’s analysis suggest that none of the ingredients is dominant and the power of the approach is essentially a “cumulative improvement from multiple changes”. This observation and conclusion are instructive and I suggest to add them to the Conclusion section.

2. Redundancy of node features: the authors used bond lengths as node features; however, the bond length is roughly the same for all residues. The effects of virtual atoms is also not clearly stated. The authors cited PiFold’s analysis to answer this question, which shows only marginal contributions by these two types of features. I suggest to add a paragraph to clearly state the effects of them.

3. Writing: The current form of the manuscript still lacks of careful proofreading. There are still many grammar errors or mistakes., e.g.

— Supplementary Table S1. ….Two-layer MLP modules were applied to replace SK modules for no-SK models

— Ref. 35 even has Chinese characters.

— Ref. 39, 44: The major words of journal/conference names are not capitalized.

**Have the authors made all data and (if applicable) computational code underlying the findings in their manuscript fully available?**

Reviewer #1: Yes

Reviewer #2: Yes

Reviewer #3: Yes

PLOS authors have the option to publish the peer review history of their article (what does this mean?). If published, this will include your full peer review and any attached files.

Reviewer #1: No

Reviewer #2: No

Reviewer #3: No

Figure Files:

Data Requirements:

Reproducibility:

References:

---

## [Editor Report · Decision Letter 2]

27 Nov 2023

Dear Dr Zhou,

We are pleased to inform you that your manuscript 'SPIN-CGNN: Improved fixed backbone protein design with contact map-based graph construction and contact graph neural network' has been provisionally accepted for publication in PLOS Computational Biology.

Best regards,

Yang Zhang

Guest Editor

PLOS Computational Biology

Nir Ben-Tal

Section Editor

PLOS Computational Biology

---

## [Editor Report · Acceptance letter]

1 Dec 2023

PCOMPBIOL-D-23-01069R2 

SPIN-CGNN: Improved fixed backbone protein design with contact map-based graph construction and contact graph neural network

Dear Dr Zhou,

I am pleased to inform you that your manuscript has been formally accepted for publication in PLOS Computational Biology. Your manuscript is now with our production department and you will be notified of the publication date in due course.

With kind regards,

Zsofia Freund
